# Caspase-mediated nuclear pore complex trimming in cell differentiation and endoplasmic reticulum stress

Ukrae H Cho[1]*, Martin W Hetzer[1,2]*

[1]Molecular and Cell Biology Laboratory, Salk Institute for Biological Studies, La Jolla, United States; [2]Institute of Science and Technology Austria (IST Austria), Klosterneuburg, Austria

**Abstract** During apoptosis, caspases degrade 8 out of ~30 nucleoporins to irreversibly demolish the nuclear pore complex. However, for poorly understood reasons, caspases are also activated during cell differentiation. Here, we show that sublethal activation of caspases during myogenesis results in the transient proteolysis of four peripheral Nups and one transmembrane Nup. 'Trimmed' NPCs become nuclear export-defective, and we identified in an unbiased manner several classes of cytoplasmic, plasma membrane, and mitochondrial proteins that rapidly accumulate in the nucleus. NPC trimming by non-apoptotic caspases was also observed in neurogenesis and endoplasmic reticulum stress. Our results suggest that caspases can reversibly modulate nuclear transport activity, which allows them to function as agents of cell differentiation and adaptation at sublethal levels.

## eLife assessment

This **important** study outlines a new role for caspases during cellular differentiation. The methodology used is **convincing** and state-of-the-art. The newly discovered cellular cascade described here uncovers that caspases can achieve high substrate specificity during differentiation. As such, the work will be of broad interest to cell biologists.

## Introduction

Caspases are a group of apoptotic proteases that target hundreds of proteins (*Araya et al., 2021*; *Mahrus et al., 2008*; *Shimbo et al., 2012*).The activation of executioner caspases, especially that of caspase-3, is equated with cell death. However, studies in the last two decades have demonstrated that it is transiently activated at sublethal levels in differentiating cells (*Burgon and Megeney, 2018*; *Connolly et al., 2014*; *Ding et al., 2016*; *Tang et al., 2015*). This phenomenon has been described in myogenic, neuronal, osteogenic, and many other types of differentiation.

The nuclear pore complex (NPC), the sole gateway between the nucleoplasm and the cytoplasm, is extensively targeted by caspase-3 (*Ferrando-May et al., 2001*; *Kihlmark et al., 2001*; *Patre et al., 2006*). A total of eight nucleoporins (Nups), the NPC subunits, are degraded by this cysteine protease: two out of the eight Nups are on the cytoplasmic side of the NPC (Nup358 and Nup214), two form NPC scaffold (Nup96 and Nup93), three reside on the nucleoplasmic side (Nup153, Tpr, and Nup50), and one is a transmembrane protein (Pom121). Hence, all major NPC substructures – cytoplasmic filaments, central scaffold, nuclear basket, and membrane-bound portion – are destroyed. Caspase-3-mediated NPC disintegration has been studied extensively in apoptotic cells, but not in differentiating cells. It is unknown if/how the NPC is modified by caspase-3 in differentiating cells. Are the same set of

**\*For correspondence:**
ucho@salk.edu (UHC);
martin.hetzer@ista.ac.at (MWH)

**Competing interest:** The authors declare that no competing interests exist.

eight Nups cleaved? What happens to nuclear transport and passive permeability barrier? How does NPC proteolysis facilitate cell identity transition?

We address these questions by examining caspase-mediated NPC proteolysis in C2C12 cells undergoing myoblast-to-myotube transition where pulse-like, non-apoptotic caspase-3 activation has been well-characterized (*Bloemberg and Quadrilatero, 2014*; *Boonstra et al., 2018*; *Fernando et al., 2002*; *Dehkordi et al., 2020*; *Murray et al., 2008*). Transient caspase-3 activation in differentiating myoblasts has been shown by immunoblotting and fluorogenic probes. Moreover, genetic deletion or pharmacological inhibition of caspase-3 significantly inhibits cell-cell fusion and expression of key muscle genes. Here, we show that during myogenesis, caspase-3 molecularly 'trims' the NPC in a reversible manner rather than dismantling the entire structure as in apoptosis. This partial degradation of the NPC transiently impairs nuclear export, but not nuclear import and passive permeability barrier, and accompanies nuclear accumulation of focal adhesion proteins, tubulins, Annexin A's, mitochondrial enzymes, etc. Of note, NPC trimming explains how FAK becomes transiently nuclear, which previously was shown to be required for myogenin de-repression (*Luo et al., 2009*). NPC trimming is observed in differentiating primary myoblasts and neural precursor cells as well, illustrating that it may be a general feature of cell differentiation. We also observed NPC trimming in ER-stressed cells, suggesting that caspase-mediated modulation of nuclear transport might represent not only a programmed step in differentiation but also a part of the cellular stress response system.

## Results

### 5 Nups are completely degraded by caspase-3 during myogenesis

Caspase-3 can degrade eight Nups (Nup358, Nup214, Nup153, Nup96, Nup93, Nup50, Tpr, and Pom121) during staurosporine-, actinomycin D-, etoposide-, or TRAIL-induced apoptosis (*Ferrando-May et al., 2001*; *Patre et al., 2006*). Recombinant caspase-3 reproduces in vivo Nup fragmentation pattern when added to permeabilized cells, and its pharmacological inhibition by DEVD-CHO blocks Nup degradation during apoptosis (*Ferrando-May et al., 2001*). Nups were identified as caspase-3 targets in multiple substrate profiling experiments as well (https://wellslab.ucsf.edu/degrabase/). Rendering them caspase-3-resistant is technically not feasible as most of them are cleaved at multiple sites – for example, 8+for Nup358 and 4+for Tpr (*Patre et al., 2006*). We observe 6+fragments for Nup153 (see below). However, human Pom121 has been made uncleavable by D531A mutation (*Kihlmark et al., 2004*).

We first asked whether caspase-3 proteolyzes Nups during myogenesis. As reported previously, we detected the active form of caspase-3 as well as that of caspase-9 and –12 in differentiating C2C12 cells (*Figure 1A* and *Figure 1—figure supplement 1A*; *Fernando et al., 2002*; *Gomez-Cavazos and Hetzer, 2015*; *Murray et al., 2008*; *Nakanishi et al., 2005*). Typically, caspase activity peaks at 16–24 hr after inducing myogenesis, and is fully quenched by 32–60 hr (see *Cell culture* in Materials and methods). As a reference, peak caspase-3 activity level in differentiating C2C12 cells is comparable to that in C2C12 cells treated with 1 μM staurosporine for 4 hr (*Murray et al., 2008*). During this time window, 4 peripheral Nups, Nup358, Nup214, Nup153, and Tpr, were reversibly and completely degraded in a caspase-dependent manner (*Figure 1—figure supplement 1B*). Complete loss of full-length proteins demonstrates that this proteolytic event takes place in the differentiating population, not just in 2–3% of the population that undergo apoptosis during C2C12 myogenesis. (We washed plates thrice with PBS before harvesting to minimize their contribution. We also show the loss of Nup fragments in differentiating cells by immunofluorescence below.) Although partially, Nup50 was proteolyzed as well (*Figure 1—figure supplement 1C*). Nup96 and Nup93, which form NPC scaffold and are cleaved during apoptosis (*Patre et al., 2006*), remained intact. (For Nup93, a faster-running, second band arises in a caspase-dependent manner (*Figure 1—figure supplement 1C*). The molecular nature of this band is unclear at the moment.) This agrees with our previous report showing the persistence of the same Nup96 and Nup93 copies through the course of myogenesis (*Toyama et al., 2019*).

For more comprehensive survey, we tested 8 additional scaffold Nups and 1 transmembrane Nup, Pom121, which is cleaved by caspase-3 in apoptotic cells (*Figure 1B*). Only Pom121 exhibited caspase-dependent proteolysis. In summary, caspase-3 degrade 4 peripheral Nups and 1 transmembrane Nup during myoblast-to-myotube transition. Given that caspase-3-exposed NPCs will lack cytoplasmic

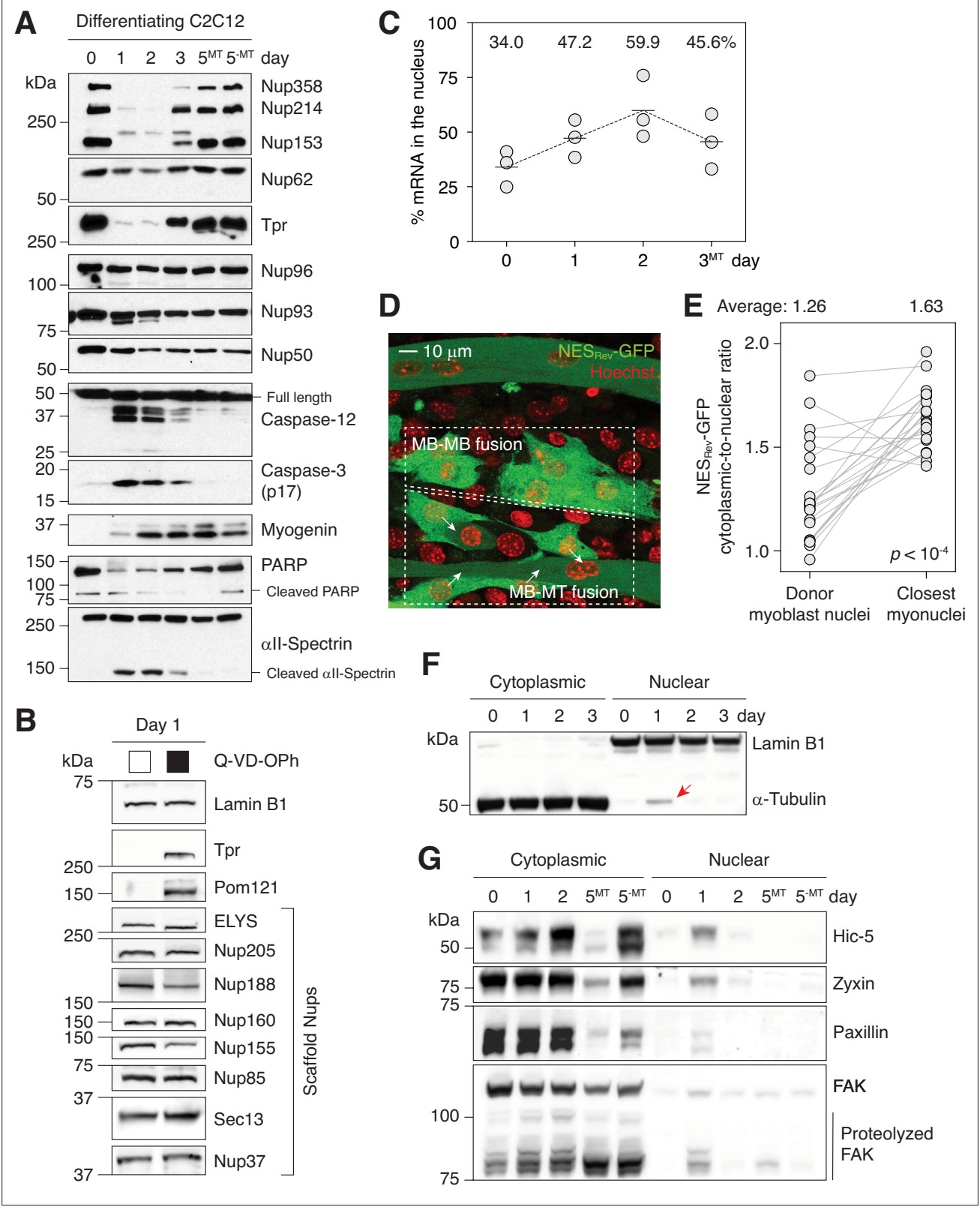

**Figure 1.** Caspases fully proteolyze 5 peripheral Nups during myogenesis. (**A**) Immunoblots showing the degradation of Nups, PARP, and αII-spectrin, the activation of caspase-3/12 and the upregulation of myogenin in differentiating C2C12 cells. (**B**) Caspase-dependent proteolysis of Nups in differentiating C2C12 cells assessed by immunoblotting. Q-VD-OPh used at 30 μM. (**C**) Nuclear-to-total mRNA ratio in C2C12 cells undergoing myogenesis. The average value of three replicates shown for each time point. (**D**) A stable C2C12 cell line that expresses NES$_{Rev}$-GFP. In dashed boxes

*Figure 1 continued on next page*

*Figure 1 continued*

are cells undergoing MB-to-MB or MB-to-MT fusion. (**E**) Cytoplasmic-to-nuclear ratio of NES$_{Rev}$-GFP was quantified for 22 donor myoblast / closest myonuclei pairs. Grey lines link each pair. (**F**) Localization of lamin B1 and α-tubulin in differentiating C2C12 cells assessed by immunoblotting. Nuclear α-tubulin marked with an arrow. Thirty μg of protein loaded per lane. (**G**) Cytoplasmic and nuclear levels of focal adhesion proteins in differentiating C2C12 cells. Twenty-five μg of protein loaded per lane. MT: myotubes, MB: myoblasts.

The online version of this article includes the following source data and figure supplement(s) for figure 1:

**Source data 1.** Unprocessed gel images.

**Figure supplement 1.** Caspase-mediated NPC trimming in differentiating C2C12 cells.

**Figure supplement 2.** Transcript levels of NPC subunits in differentiating C2C12 cells.

**Figure supplement 3.** Nuclear export impairment in differentiating C2C12 cells.

**Figure supplement 4.** Nuclear retention of NES-containing proteins in differentiating C2C12 cells.

**Figure supplement 5.** Proximity ligation assay.

filaments (Nup358 and Nup214), nuclear basket (Nup153 and Tpr), and the C-terminal tail of Pom121 that extends into the cytoplasm, we termed this phenomenon "NPC trimming". Importantly, since the NPC core is unaffected, functional NPCs can be quickly restored once caspases are quenched. RNA-seq shows that the transcript levels of all but one Nups remain stable (*Figure 1—figure supplement 2*). Nup210 is the only exception whose expression level changes dramatically during myotube formation at both transcript and protein levels as we have previously reported (*D'Angelo et al., 2012*).

Calpains are another class of proteases that are activated in differentiating myoblasts (*Louis et al., 2008*), and are known to proteolyze Nups in multiple contexts (*Bano et al., 2010*; *Sugiyama et al., 2017*; *Yamashita et al., 2017*). We tested whether calpain inhibition prevents the digestion of Nups (*Figure 1—figure supplement 1B*). Interestingly, it delayed but did not prevent caspase-3/12 activation and peripheral Nup degradation. This suggests that they participate in caspase activation process but are not responsible for NPC trimming.

Given that central FG Nups, such as Nup98 and Nup62, are not targeted by caspases (*Ferrando-May et al., 2001*; *Patre et al., 2006*), we surmised that the nuclear passive permeability barrier would remain intact. Indeed, nuclear exclusion of ≥40 kDa dextrans demonstrates that the passive barrier is functional even when the NPC is partially disintegrated (*Figure 1—figure supplement 1D*).

## Caspase-mediated NPC trimming impairs nuclear export

The loss of peripheral Nups suggested that the engagement of nuclear transport receptors, and thus nuclear transport, might be affected. The removal of Nup358, Nup214, Nup153, or Tpr has been shown to block nuclear export and cause nuclear accumulation of RNAs and NES (nuclear export signal)-containing proteins (*Aksenova et al., 2020*; *Forler et al., 2004*; *Hutten and Kehlenbach, 2006*; *Ullman et al., 1999*). We thus assessed mRNA export during the first three days of myogenic differentiation by combining nuclear isolation and oligo(dT) bead-based poly(A)$^+$ RNA purification. The nuclear-to-total mRNA ratio increases from 34% to 60% during differentiation, indicating an impairment in mRNA export (*Figure 1C* and *Figure 1—figure supplement 3A–B*). Of note, the distribution of housekeeping gene RNAs, 18 S rRNA and *Gapdh* transcript, remains unchanged during this period as they have extended half-lives (*Figure 1—figure supplement 3C*; *Lee et al., 2010*; *Yi et al., 1999*). This might explain how cells can continue to synthesize essential proteins and maintain homeostasis even when RNA export is temporarily inhibited.

We then examined Crm1-dependent nuclear export of proteins. The cytoplasmic-to-nuclear ratio of NES$_{Rev}$-GFP, a widely used export probe, was significantly lower in myoblasts undergoing cell-cell fusion than in myotubes (*Figure 1D and E*, and *Figure 1—figure supplement 3D*). Similarly, the cytoplasmic-to-nuclear ratio of NES$_{PKI}$-GFP significantly decreased when myoblasts started to produce "bubbling" blebs and became fusion-competent (*Figure 1—figure supplement 3E–G*; *Lian et al., 2020*). We also asked if the localization of α-tubulin is affected. It is normally absent in the nucleus due to the presence of multiple NESs (*Schwarzerová et al., 2019*). Strikingly, 1 day post-differentiation when NPCs are partially disintegrated, α-tubulin was detected in the nuclear lysate, although at a relatively low level (*Figure 1F*). When NPC trimming was blocked by Q-VD-OPh, a pan-caspase inhibitor, its nuclear translocation was suppressed (*Figure 1—figure supplement 4A*). We next checked if focal adhesion proteins behave similarly since they (1) contain NESs as well and (2) are likely to be released

to the cytoplasm by calpains in differentiating C2C12 cells (*Louis et al., 2008*). Remarkably, all four focal adhesion proteins that we examined (Hic-5, zyxin, paxillin, and FAK) became transiently nuclear during myogenesis (*Figure 1G*). In the presence of Q-VD-OPh, their nuclear entrapment was notably suppressed, albeit not fully (*Figure 1—figure supplement 4B*). The residual amount can be attributed to forced import during myogenesis by their binding partners that make use of them as transcription cofactors in the nucleus. We indeed detect interaction between FAK and MBD2 (*Luo et al., 2009*) and Hic-5 and androgen receptor (*Alpha et al., 2020*) by proximity ligation assay (*Figure 1—figure supplement 5*).

In summary, differentiation-associated caspase activity impairs nuclear export and leads to the nuclear retention of NES-containing proteins. (Nuclear import is discussed below.) Given the rapid influx of NES-harboring proteins, we hypothesized that leptomycin B, a Crm1 inhibitor, could support myogenesis. Indeed, when differentiation medium is supplemented with a partial dose of leptomycin B, myogenin expression is enhanced (*Figure 1—figure supplement 4C*). This agrees with the reported ~twofold myogenin enhancement by 6 hr of leptomycin B pre-treatment in growth medium (*Bustos et al., 2015*). In contrast, a full dose of Q-VD-OPh in differentiation medium significantly suppresses myogenin upregulation (*Figure 1—figure supplement 4D*). Q-VD-OPh significantly reduces cell-cell fusion and myosin heavy chain expression as well (*Murray et al., 2008*).

## Tpr and Pom121 C-terminal fragments dissociate from the NPC

Since Nups are large proteins that are held in place via multiple interactions, we asked whether the entire proteins are released from the NPC after proteolysis. Using four different antibodies (*Figure 2—figure supplement 1A*), we first analyzed whether various domains of Nup153 and Tpr are retained in late-stage apoptotic nuclei where caspase levels are significantly higher than in differentiating cells and chromatins are fully condensed. Immunostaining revealed that only the C-terminal epitope of Tpr dissociates from the NPC while proteolyzed Nup153 and Tpr N-terminus remain bound (*Figure 2A and B*). Their persistent association with the NPC is reminiscent of that of enterovirus 2 A or rhinovirus 3 C protease-cleaved Nup98 (*Park et al., 2015*; *Walker et al., 2013*).

Differentiating myoblasts exhibited a proteolytic phenotype that is distinct from that of apoptotic cells. Quantitative analysis suggests that Tpr C-terminal fragment dissociates partially (*Figure 2C and D*). Only in 5–10% of the cells were the fragment completely dissociated as in apoptotic cells (dotted boxes in *Figure 2C*). 'TprC⁻' population can be clearly distinguished from dying cells by nuclear morphology (*Figure 2—figure supplement 1B*). Moreover, while the nucleoplasmic pool of Crm1 is lost and it becomes confined to the nuclear periphery in apoptotic cells (*Figure 2—figure supplement 1C*), Crm1 localization did not change in differentiating cells, even in TprC⁻ cells (*Figure 2C*). This might explain why NPC trimming results in a subdued nuclear export defect compared to a full dose of leptomycin B; NES-containing focal adhesion proteins accumulate in the nucleus to a limited extent during NPC trimming (*Figure 1G*) but become predominantly nuclear upon leptomycin B treatment (*Nix et al., 2001*; *Ossovskaya et al., 2008*).

To further validate (in)complete retention of Tpr C-terminal tail and Nup153 fragments, respectively, we isolated the nuclei of differentiating C2C12 cells and fractionated them into 'nuclear insoluble' (nuclear envelope, NPC, and chromatin-bound) and 'nuclear soluble' (*Figure 2E*; fractionation steps detailed below). Nup153 fragments were found exclusively in the former. In contrast, the C-terminal piece of Tpr, which is expected to be ~37 kDa (*Figure 2—figure supplement 1A*; *Ferrando-May et al., 2001*), was detected in both subfractions. This agrees with the partial-dissociation-upon-cleavage model.

We also analyzed Pom121 fragmentation. Pom121 has a short transmembrane and a long pore-side domain (aminoacid 35–55 and 56–1249 respectively in human; *Figure 2—figure supplement 1A*). The latter is caspase-cleaved and is undetectable by immunostaining in apoptotic cells (*Kihlmark et al., 2001*; *Kihlmark et al., 2004*). Although Pom121 is proteolyzed to completion at 24 hr post-differentiation (*Figure 1B*), immunostaining shows that its pore side domain does not fully dissociate from the NPC (*Figure 2F and G*). As in the case of Tpr, Pom121 C-terminus immunoreactivity does decrease on average but only a subset of nuclei exhibits its full loss. To conclude, there is some heterogeneity in the behavior of caspase-generated Nup fragments while NES-GFP reporter demonstrates that most, if not all, NPCs become export-defective. Lastly, we confirmed that trimmed NPCs do not cluster like they do in late-stage apoptosis via maximum intensity Z projection (*Figure 2—figure*

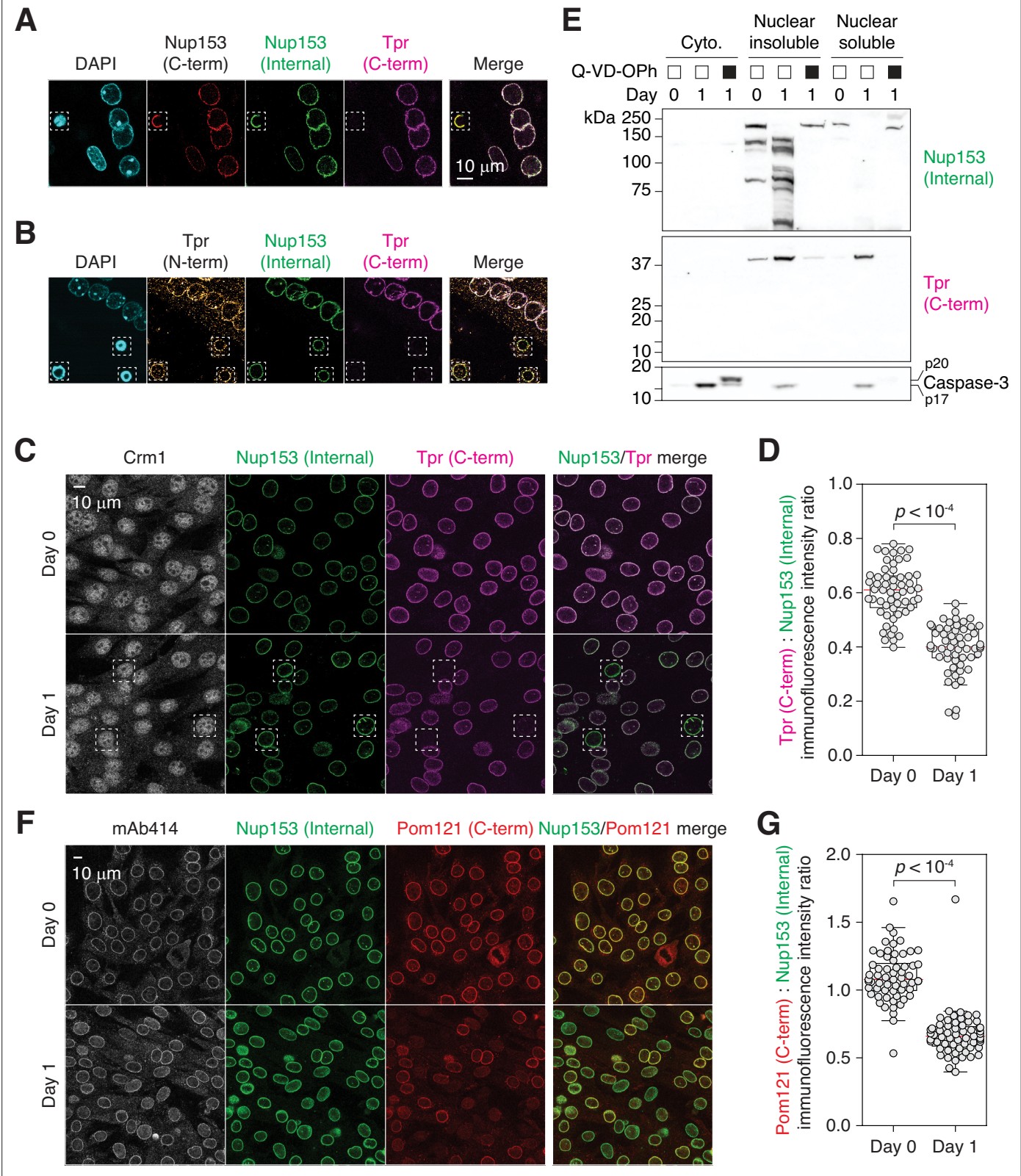

**Figure 2.** Tpr and Pom121 C-terminal fragments dissociate from the NPC during myogenesis. (**A and B**) C2C12 cells immunostained using two Nup153 and two Tpr antibodies. In dashed boxes are apoptotic nuclei. (**C**) C2C12 cells undergoing myogenesis immunostained for Crm1, Nup153, and Tpr. In dashed boxes are TprC⁻ cells. (**D**) Tpr C-term immunoreactivity (normalized against Nup153 internal domain immunoreactivity) determined for 50+nuclei. (**E**) Nup153, Tpr, and caspase-3 fragments in subcellular fractions of differentiating C2C12 cells visualized by immunoblotting. Q-VD-OPh

*Figure 2 continued on next page*

*Figure 2 continued*

used at 30 µM. (**F**) C2C12 cells undergoing myogenesis immunostained for mAb414, Nup153, and Pom121. (**G**) Pom121 C-term immunoreactivity (normalized against Nup153 internal domain immunoreactivity) determined for 50+nuclei.

The online version of this article includes the following source data and figure supplement(s) for figure 2:

**Source data 1.** Unprocessed gel images.

**Figure supplement 1.** Immunofluorescence staining of apoptotic and differentiating C2C12 cells.

**Figure supplement 2.** Immunofluorescence staining of differentiating C2C12 cells.

*supplement 1D*; *Kihlmark et al., 2001*), and that the localization of importins is unaffected by NPC trimming (*Figure 2—figure supplement 2*).

## NPC trimming coincides with a rapid change in the nuclear proteome

Next, we employed quantitative mass spectrometry to study how NPC trimming affects the nuclear proteome. For better coverage of low-abundance proteins, we fractionated nuclei into 'nuclear insoluble' and 'nuclear soluble' by hypotonic lysis (*Figure 3A*). The former contains proteins strongly bound to the NPC, nuclear envelope, and chromatin, whereas in the latter are ones solubilized in the nucleoplasm (*Figure 3B*). Using tandem mass tag labeling, we determined how the levels of ~2800 proteins in the two subfractions change during caspase activation and NPC trimming, that is, before and after 24 hr of myogenic differentiation. We overlaid RNA-seq data on top of mass spectrometry result and generated a color-mapped volcano plot for the 'nuclear soluble' fraction (*Figure 3C–D* and *Supplementary file 1*). Our multi-omic analysis well matches previous candidate-based studies. As shown in *Figure 3C*, alpha-actinin 2 (Actn2) is known to be transcriptionally upregulated and removed from the nucleus during myogenesis (*Lin et al., 2010*).

In the 'nuclear soluble' fraction, there were 236 proteins that increased >fourfold in the first 24 hr. We grouped them by function/localization (*Figure 4A*). All focal adhesion proteins but tensins became significantly more nuclear as expected. Several α/β-tubulin isoforms and microtubule regulators displayed similar behavior. Interestingly, Tppp3 (Tubulin polymerization promoting protein family member 3), which has been reported to localize to nucleolar fibrillar center (the Human Protein Atlas; HPA047629), was the top hit in this group. Another noticeable group was Annexin A's. Among them is Annexin A2, which is an NES-containing protein that becomes more nuclear in genotoxic condition and mitigates DNA damage. The nuclear level of S100a10, an Annexin A2 interactor, increased sharply as well (*Figure 3D*). We also find that several mitochondrial enzymes, but not ATP synthase or TIM/TOM complex subunits barring Tomm34, accumulate in the nucleus. A subset of the Krebs cycle enzymes has been reported to enter the nucleus during embryogenesis (*Nagaraj et al., 2017*), and we find the same in myogenic differentiation. Lastly, the nuclear levels of most E3 ligases and proteasome subunits remain relatively stable, but there were a few exceptions such as Itch, Ufl1, Psme1, and Psmb3.

In addition to caspase-mediated NPC trimming, other mechanisms could increase the nuclear level of a protein during myogenesis. Our color-mapped volcano plot shows that most transcriptionally upregulated proteins (red data points in *Figure 3C and D*) become more abundant in the nucleus, demonstrating that enhanced synthesis is another major factor. Enhanced nuclear import and/or stability can be involved as well. To determine the contribution of NPC trimming to a given translocation event, one would need to generate five caspase-resistant Nups, which for technical reasons is impractical. Our alternative approach was to pharmacologically inhibit caspases during differentiation and test if the influx/retention of non-nuclear proteins is mitigated. Out of 700 proteins that show >twofold increase in the nuclear level during the first 24 hr (area shaded in red in *Figure 4B*), all but four display less increase when caspases are inhibited. This is in line with our view that NPC trimming plays a key role in shaping the nuclear proteome (*Figure 4C*). In sum, our data (1) illustrates that the nuclear proteome changes dramatically and rapidly in response to the myogenic signal, (2) reveals the importance of protein translocation in cell differentiation, and (3) provides a fresh list of proteins that have been overlooked in the context of myotube formation.

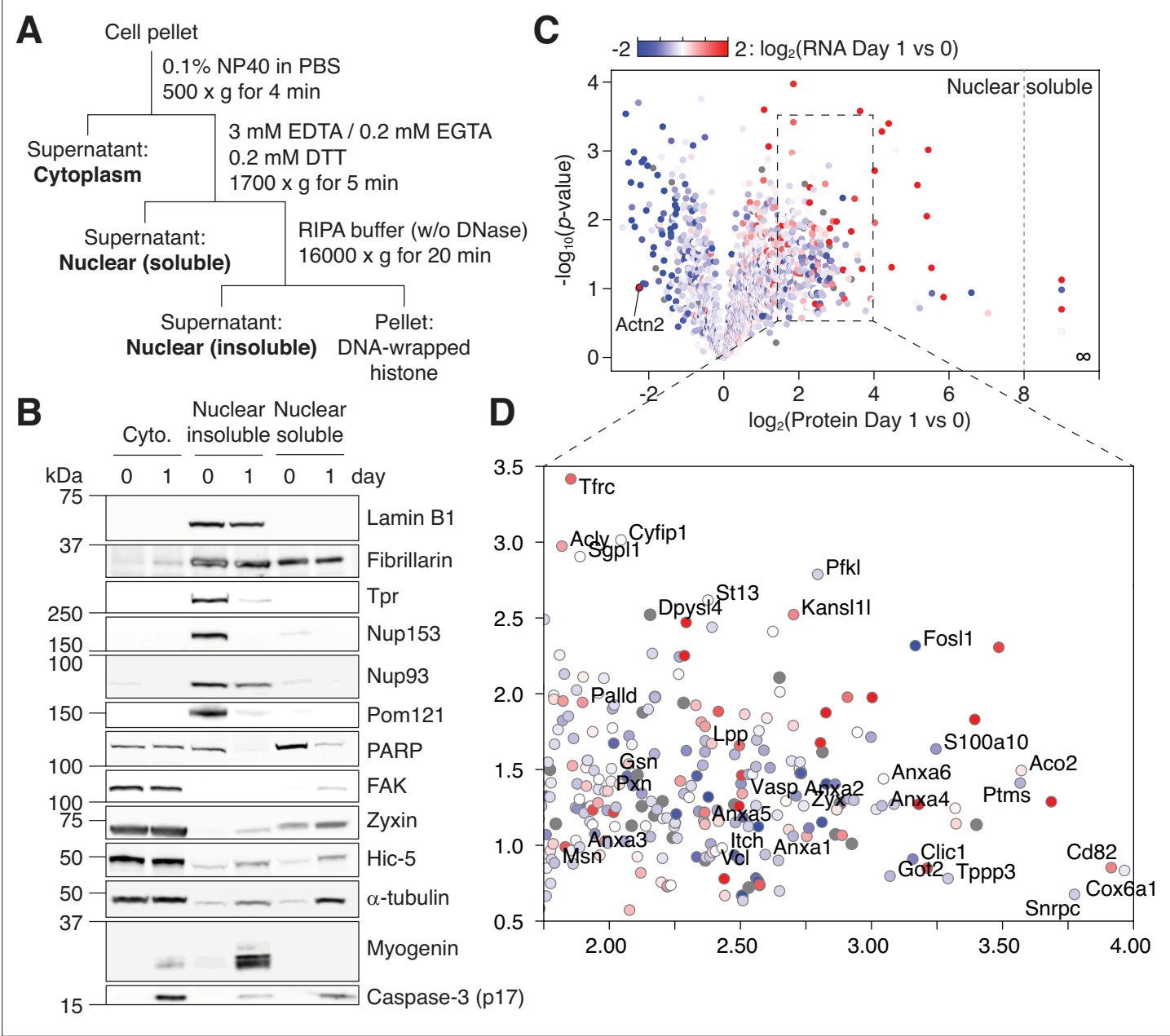

**Figure 3.** NPC trimming accompanies nuclear accumulation of cytoplasmic, plasma membrane, and mitochondrial proteins. (**A**) Preparation of subcellular fractions for quantitative mass spectrometry. (**B**) Subcellular fractions were validated using multiple protein markers. (**C**) Volcano plot showing how protein levels in the 'nuclear soluble' fraction change during the first 24 hr of myogenic differentiation. Each data point colored to represent their transcriptional change during the same time frame. (**D**) Zoomed-in view of the dotted square in (**C**).

The online version of this article includes the following source data for figure 3:

**Source data 1.** Unprocessed gel images.

## NPC trimming occurs after survivin downregulation and before myogenin upregulation

We find that caspase-mediated NPC trimming (*Figure 5A*) coincides with a rapid nuclear proteome change. We next wanted to pinpoint the timing of this event. Cell cycle exit is one of the earliest events in myogenesis. Survivin is a chromosomal passenger complex subunit that is highly expressed in G2/M phase but rapidly downregulated in early G1 (*Li et al., 1998*). In differentiating C2C12 cells, survivin downregulation precedes caspase-3 activation, indicating that the latter occurs after the very

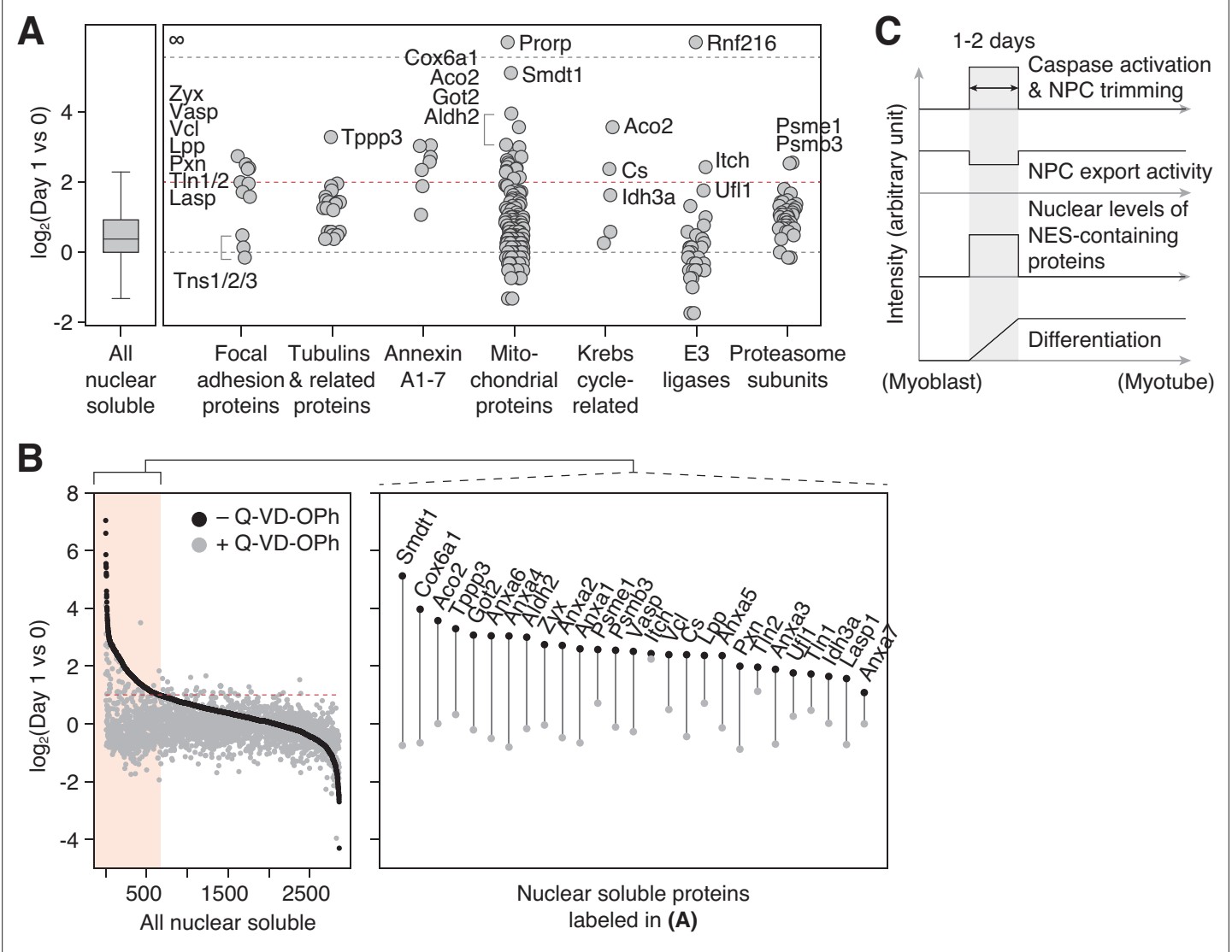

**Figure 4.** NPC trimming accompanies nuclear accumulation of cytoplasmic, plasma membrane, and mitochondrial proteins. (**A**) Protein groups detected in the 'nuclear soluble' fraction. Red dotted line: y=2. (**B**) Protein level change in the 'nuclear soluble' fraction when C2C12 cells are differentiated with or without 30 μM Q-VD-OPh. Red dotted line: y=1. Data points for proteins labeled in (**A**) are shown in the right box. (**C**) Visual summary of multiple events that take place during myoblast-to-myotube transition.

last cell division and as cells enter G0 (*Figure 5B*). We then determined the timing of caspase-3 activation relative to the upregulation of myogenin, which marks the point-of-no-return. Both immunoblotting and immunstaining show that active caspase-3 arises before myogenin (*Figure 5B* and *Figure 5—figure supplement 1A*). Interestingly, cells can remain NPC-trimmed for a brief period after myogenin upregulation. We not only identified TprC⁻ Myog⁻ but also TprC⁻ Myog⁺ cells (*Figure 5—figure supplement 1A*). The presence of the latter signifies that nuclear import is functional even when NPCs are trimmed. The same transitory cell states appear in differentiating primary myoblasts (*Figure 5—figure supplement 1B*), demonstrating that this is not a C2C12-specific phenomenon. Caspase-3 is quenched by the time myosin heavy chain is expressed, that is, when cell-cell fusion occurs (*Figure 5—figure supplement 1C*). When myogenesis is accelerated by GFP-myogenin overexpression, caspase-3 is quenched a day earlier compared to the control condition (*Figure 5—figure supplement 1C*), signifying that a caspase-quenching pathway exists and is activated once myoblasts pass a certain differentiation stage. *Figure 5C* summarizes the order of mentioned events.

We also checked whether caspase-3 activation, and hence NPC trimming, is affected by key signaling pathways involved in myogenesis: ERK1/2, calcineurin/NFAT, and Notch. ERK1/2 inhibition

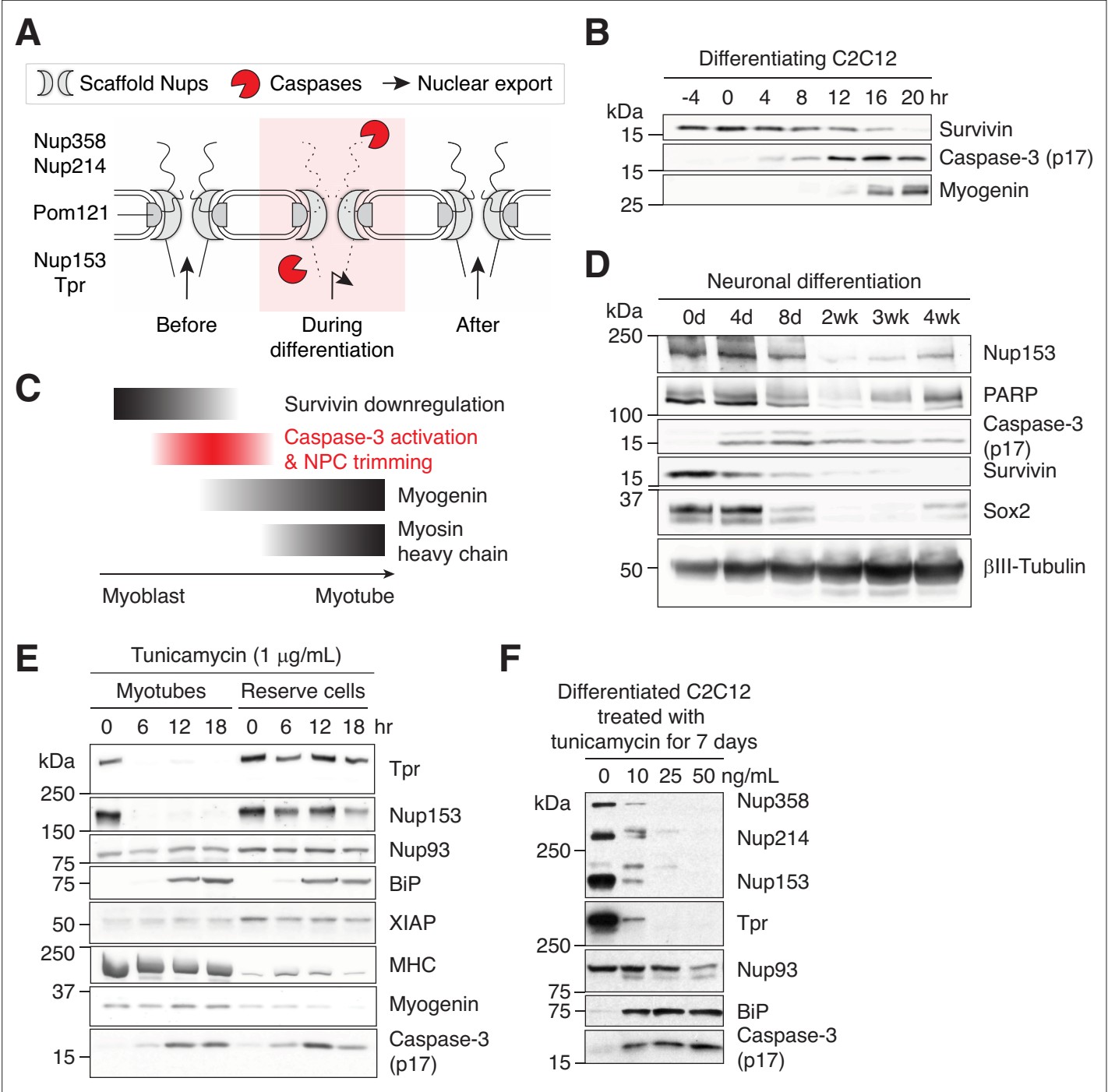

**Figure 5.** Caspase-mediated NPC trimming occurs during neurogenesis and ER stress. (**A**) Schematic representation of caspase-mediated NPC trimming. (**B**) Down-/up-regulation of survivin, caspase-3, and myogenin in differentiating C2C12 cells. (**C**) Relative timing of key events in myogenesis. (**D**) Expression levels of Nup153, PARP, active caspase-3, survivin, and neuronal differentiation markers (Sox2 and βIII-tubulin) were determined in differentiating neurons. (**E**) Acute ER stress induced using 1 μg/mL tunicamycin in C2C12 myotubes and reserve cells. The levels of Nups, BiP, XIAP, myosin heavy chain (MHC), myogenin, and active caspase-3 levels were monitored by immunoblotting. (**F**) Chronic ER stress was induced in differentiated C2C12 cells for 7 days using low doses of tunicamycin. Nups, BiP, and active caspase-3 levels were assessed.

The online version of this article includes the following source data and figure supplement(s) for figure 5:

**Source data 1.** Unprocessed gel images.

**Figure supplement 1.** Activation of caspases and proteolysis of their substrates.

stimulates myogenesis by promoting cell-cycle exit as well as cell-cell fusion (*Eigler et al., 2021*). Consistently, we find that SCH772984, an ERK1/2 inhibitor, accelerates survivin downregulation, caspase-3 activation, Nup153 degradation, and myogenin induction (*Figure 5—figure supplement 1D*). When the nuclear translocation of NFAT was blocked by FK506, a calcineurin inhibitor, cells failed to upregulate myogenin as previously reported (*Armand et al., 2008*) but still activated caspase-3 (*Figure 5—figure supplement 1E*). This result illustrates that the calcineurin/NFAT pathway is orthogonal to the caspase-3 activation pathway. Lastly, DAPT, a γ-secretase inhibitor that blocks Notch intracellular domain release, neither affected myogenin upregulation nor caspase-3 activation (*Figure 5—figure supplement 1F*).

## NPC trimming occurs during neurogenesis and ER stress

We next asked whether caspase-mediated NPC trimming is an event that is specific to myogenesis, which results in huge, multinucleated cells, or represents a general phenomenon associated with cell differentiation. To test this, neural precursor cells were differentiated into post-mitotic neurons over 4 weeks, and caspase-related events were analyzed (*Figure 5D*). As in myogenic differentiation, survivin is highly expressed in precursor cells but undetectable in mature neurons, and the activation of caspase-3 and the proteolysis of Nup153 and PARP occurred concomitantly. This suggests that caspases play similar roles during neuronal differentiation, albeit over a longer duration.

Our results so far suggest that NPC trimming and resultant nuclear proteome change are parts of a well-orchestrated sequence of events during early stages of cell differentiation. Interestingly, however, the calpain/caspase activation cascade in differentiating myoblasts is reminiscent of that in unfolded protein response. We thus asked whether ER stress alone can trigger partial NPC disintegration. C2C12 myotubes and reserve cells (mononucleated population) were treated with 1 μg/mL tunicamycin (*Figure 5E*). Active caspase-3 arose in both cell types at comparable levels; however, in myotubes, Nup153 and Tpr were fully proteolyzed by 6 hours. The discrepancy can be attributed to XIAP. XIAP is the major caspase-inhibiting protein, and its level is significantly lower in myotubes at 0 hr. (XIAP in myotubes migrates faster than that in reserve cells during gel electrophoresis, possibly due to Ser87 dephosphorylation, which stabilizes XIAP *Kato et al., 2011*.) ER stress-induced NPC trimming exhibits multiple similarities to that during myogenesis: (1) NES$_{Rev}$-GFP becomes more nuclear (*Figure 5—figure supplement 1G*), (2) 5–10% of myotubes become TprC⁻ while retaining Nup153 epitopes (*Figure 5—figure supplement 1H*), and (3) Nup93 remains unaffected, again illustrating that scaffold Nups are better protected from caspases (*Figure 5E*). Additionally, we tested whether low-level but sustained ER stress induces the same phenotype since tunicamycin treatment at the full dose may not be physiologically relevant. We maintained differentiated C2C12 cells at low doses of tunicamycin (<50 ng/mL) for a week (*Figure 5F*). As in differentiating myoblasts and acutely ER stressed-myotubes, the breakdown of Nup93 was minimal, whereas 4 peripheral Nups

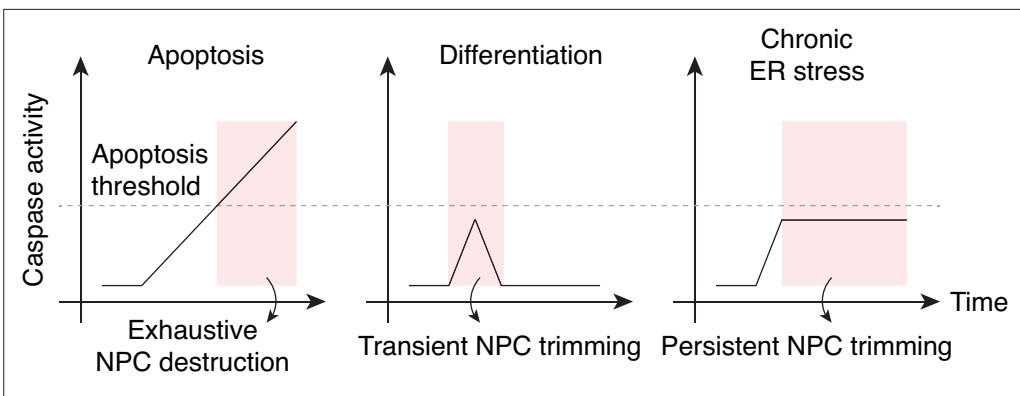

**Figure 6.** Caspase activation and NPC proteolysis patterns in apoptosis, cell differentiation, and chronic ER stress.

The online version of this article includes the following figure supplement(s) for figure 6:

**Figure supplement 1.** Transient nuclear accumulation of FAK resets MBD2-mediated genome regulation during myogenesis.

were noticeably degraded. In short, chronic ER stress forces the NPC to remain in the trimmed form for a prolonged period before apoptosis is triggered.

## Discussion

During apoptosis, the NPC is completely demolished by fully activated caspase-3. In differentiating cells, caspase-3 activation is sublethal, and it selectively targets NPC projections, that is, cytoplasmic filaments, nuclear basket, and Pom121 pore-side domain (*Figure 6*). Caspase-mediated NPC trimming is a reversible process that transiently impairs nuclear export. This phenotype is somewhat reminiscent of that of cells exposed to picornavirus 2 A protease, where Nup98 is selectively degraded from the NPC and mRNA/protein export is inhibited while nuclear import and passive permeability barrier remain mostly functional (*Serganov et al., 2022*).

NPC trimming coincides with the nuclear influx/retention of non-nuclear proteins. Why do differentiating cells need non-nuclear proteins in the nucleus? While it will require multiple follow-up studies to answer this question, NPC trimming provides an explanation to one key event in myogenesis whose most upstream trigger remained unknown. The nuclear level of FAK, a focal adhesion protein with both nuclear export and localization signals, was previously shown to surge during myogenesis to remove MBD2 – the main component of the gene-repressing NuRD (nucleosome remodeling and deacetylation) complex – from methylated CpGs within the *Myog* promoter (*Figure 6—figure supplement 1A*; *Luo et al., 2009*). In other words, nuclear FAK de-represses *Myog*. Nevertheless, how FAK rapidly translocates into the nucleus in differentiation myoblasts has remained unclear. We here show that caspase-mediated NPC trimming impairs its export from the nucleus during myogenic differentiation. In line with this model, we find that the loss of MBD2-genome binding occurs within 24 hr as FAK becomes nuclear and precedes the degradation of MBD2 protein itself during myogenesis (*Figure 6—figure supplement 1B and C*). Of note, in mechanically stressed cells, it is well-established that focal adhesion proteins quickly translocate to the nucleus to function as transcription co-factors and induce rapid gene expression changes necessary for stress resolution (*Hervy et al., 2006*). It is possible that this modus operandi has been adapted for cell differentiation.

In addition, we find in the nucleus diverse classes of proteins that have been linked to DNA damage response, such as Annexin A2, E3 ligases (Itch and Ufl1), and proteasome subunits (Psmb3) (*Chang et al., 2019*; *Jacquemont and Taniguchi, 2007*; *Madureira et al., 2012*; *Qin et al., 2019*). This is in line with the fact that DNA damage/repair is required for myogenesis (*Al-Khalaf et al., 2016*). Multiple mitochondrial proteins enter the nucleus as well, hinting that nuclear-mitochondrial communication plays a role myogenic differentiation (*Soledad et al., 2019*).

NPC trimming might also be anNPC-genome interaction reset mechanism. It is well established that NPC-genome interaction (1) is cell type-specific, (2) is reconfigured during differentiation, and (3) regulates cell identity genes (*Cho and Hetzer, 2020*; *Ibarra et al., 2016*; *Jacinto et al., 2015*; *Labade et al., 2021*; *Liang et al., 2013*). Transient proteolysis of the nuclear basket can be an elegant reinitialization mechanism. This is an interesting hypothesis to test in the future.

In the context of acute ER stress, NPC trimming might be a stress-comping mechanism that curtails mRNA export and reduces the protein synthesis/folding load. However, if ER stress becomes chronic and caspases linger for a prolonged period, it can jeopardize cellular homeostasis as nuclear levels of NES-containing proteins will remain abnormally high. Our quantitative, subcellular fraction mass spectrometry workflow (*Figure 3A*) can be employed to examine the nuclear proteome change during chronic ER stress.

The present study establishes caspase-mediated NPC trimming as a key event in cell differentiation and stress response. It also points to the idea that exhaustive NPC disintegration during apoptosis may simply be an extreme version of this phenomenon. In fact, some consider apoptosis as a radical form of caspase-mediated cellular transformation (*Burgon and Megeney, 2018*; *Connolly et al., 2014*), and our results suggest that time is ripe to reassess the death-centric view of caspases.

## Materials and methods

### Cell lines

C2C12 cells were obtained from ATCC, and HEK293T cells, from the Salk Stem Cell Core Facility. The identity of both cell lines has been authenticated by ATCC, and we have confirmed that both cell lines are negative for mycoplasma. H9 embryonic stem cells, from the Gage lab (the Salk Institute); and mouse primary myoblasts, from the Alessandra Sacco lab (Sanford Burnham Prebys).

### Immunoblot techniques

Cultured cells were washed with phosphate-buffered saline (PBS) and harvested by trypsinization. Cell pellet was washed with PBS, and lysed in RIPA buffer (50 mM Tris-HCl pH 7.5, 150 mM NaCl, 1% Triton X100, 0.5% sodium deoxycholate, 0.1% sodium dodecyl sulfate) containing protease and phosphatase inhibitors (Pierce Protease Inhibitor Mini Tablet, EDTA-free, Thermo Scientific; PhosSTOP, Roche) for 45 min at 4 °C. Insoluble material was pelleted at 16,000 x $g$ at 4 °C for 20 min. Fifteen to –30 µg of total protein per sample were added 6 x Laemmli sample buffer, boiled for 4 min, and loaded on a Tris or Bis-tris gel for electrophoresis. Proteins were transferred to a nitrocellulose membrane and stained with Ponceau S solution to confirm equal protein loading and successful transfer (**Source Data File 1**). After washing with Tris-buffered saline containing 0.05% (w/v) Tween 20 (TBST) several times, the membrane was blocked with 5% non-fat milk power in TBST for an hour at room temperature, and subsequently immunoblotted at 4 °C overnight with primary antibodies listed in *Supplementary file 2a*. Chemiluminescent detection was conducted using either SuperSignal West Pico or Femto kits (Thermo Scientific) after 45-min incubation with secondary antibodies listed in *Supplementary file 2b* at room temperature. Western blot images were obtained using KwikQuant Imager (Kindle Biosciences) or Odyssey CLx (LI-COR).

### Cloning

In-fusion cloning was performed using the In-Fusion HD EcoDry Cloning Plus kit (Takara), and standard cut-and-paste cloning using T7 DNA ligase (New England Biolabs). NES$_{Rev}$: LPPLERLTL. NES$_{PKI}$: LALK-LAGLDI. See *Supplementary file 2c* for more details.

### Cell culture

C2C12 cells were cultured in Dulbecco's Modified Eagle Medium (DMEM) supplemented with 20% fetal bovine serum and penicillin-streptomycin. For myogenic differentiation, they were grown to confluency, washed with PBS twice, and added DMEM with 2% horse serum and the same antibiotics. C2C12 differentiation medium was replenished every 24 or 48 hr. After 120 hr, mature myotubes and reserve cells were obtained. Myotubes were harvested with minimal contamination of reserve cells by mild trypsinization (1:3 or 1:4 dilution of trypsin in PBS). We want to note that the exact timing and duration of caspase activation, and thus NPC trimming, varies depending on cell confluency, culture substrate, growth area-to-medium volume ratio, passage number, etc. Compare *Figure 1A* and *Figure 1—figure supplement 1B*. Typically, caspase activity peaks at 16–24 hr after inducing myogenesis, and is fully quenched by 32–60 hr in differentiating C2C12 cells. H9 embryonic stem cells were differentiated to neural precursor cells using a previously published method (*Marchetto et al., 2017*). Purified neural precursor cells were cultured in neurogenic conditions (DMEM/F12 based medium with 1x N2, 1 x B27, 20 ng/mL GDNF, 20 ng/mL BDNF, 1 mM cAMP, and 200 nM ascorbic acid) for 4 weeks to generate mature post-mitotic neurons.

### C2C12 cell fractionation

Cells were harvested by trypsinization, washed with PBS, and chilled on ice. Cells were then lysed in ice-cold 0.1% NP40 in PBS, and rotated at 4 °C for 10 min. Nuclei were pelleted by centrifugation at 500 x $g$ at 4 °C for 4 min, and the supernatant (cytoplasmic fraction) was transferred to a fresh tube and stored at –80 °C until use. Nuclei were again resuspended in ice-cold 0.1% NP40 in PBS, rotated at 4 °C for 5 min, and pelleted at 500 x $g$ at 4 °C for 4 min. The supernatant was discarded, and the pellet (nuclear fraction) was stored at –80 °C until use or further fractionated into 'nuclear soluble' and 'nuclear insoluble'. To obtain the former, nuclei were added 3 mM EDTA, 0.2 mM EGTA, and 0.2 mM DTT, and rotated at 4 °C for 45 min. Hypotonically lysed nuclei were centrifuged at 1700 x $g$ at 4 °C

for 5 min, and the supernatant was transferred to a fresh tube ('nuclear soluble'). Lysed nuclei were washed once with the same buffer, and then further extracted using RIPA buffer at 4 °C for 45 min. DNA-wrapped histones were pelleted at 16,000 x *g* at 4 °C for 20 min, and the supernatant was transferred to a fresh tube ('nuclear insoluble').

## Lentivirus packaging, infection, and selection
Third-generation lentiviral protocol was followed to produced virus in HEK293T cells. C2C12 cells were infected with viral supernatant at 30–40% confluency in the presence of 6 µg/mL polybrene for 24 hours, and selected 24 hr after infection with 1 mg/mL puromycin.

## RNA fluorescence in situ hybridization (FISH)
C2C12 cells were grown on a No. 1.5 coverslip placed in a 12-well cell culture plate. After removing media, cells were washed with PBS, and fixed in 3.7% paraformaldehyde in PBS for 10 min at room temperature. Fixed cells were washed with PBS twice and permeabilized in 70% (vol/vol) ethanol at 4 °C for at least a day. We then followed Stellaris RNA FISH protocol for adherent cells (https://www.biosearchtech.com/support/resources/stellaris-protocols) to fluorescently visualize RNAs. Fluorescent images were obtained using a Leica SP8 confocal microscope equipped with a 63 x oil-immersion objective. Images were cropped and pseudocolored using FIJI. See *Supplementary file 2d* for RNA FISH probes used in this study.

## Immunofluorescence
C2C12 cells and mouse primary myoblasts were grown on a chambered cell culture slide (Ibidi). For the latter, slides were coated with Matrigel prior to seeding. Cells were fixed in PBS containing 2% paraformaldehyde for 10 min at room temperature, and washed with PBS 3 times. Fixed cells were permeabilized and blocked in immunofluorescence buffer (PBS containing 0.1% Triton-X, 0.02% sodium dodecyl sulfate, and 10 mg/mL bovine serum albumin) for 30 min. The cells were then incubated with primary antibodies diluted in immunofluorescence buffer at room temperature for 2 hr, washed three times with immunofluorescence buffer, and incubated with the appropriate secondary antibodies diluted in immunofluorescence buffer at room temperature for 45 min. Finally, the cells were washed with immunofluorescence buffer three times, and added VECTASHIELD antifade mounting medium with DAPI (Vector Laboratories). Fluorescent images were obtained using a Leica SP8 confocal microscope equipped with a 63 x oil-immersion objective. Images were cropped and pseudocolored using FIJI. See *Supplementary file 2a and b* for antibodies used for immunofluorescence.

## Poly(A)$^+$ RNA quantification
Poly(A)$^+$ RNA was isolated from nuclei and whole cells (see C2C12 cell fractionation above) using the Magnetic mRNA Isolation Kit (New England Biolabs). Two rounds of binding, washing, and elution were performed. Eluted RNA was quantified via Qubit RNA HS Assay (Invitrogen).

## Proximity ligation assay
Proximity ligation assay was performed using a Duolink in situ Red Starter Kit Mouse/Rabbit (Sigma). Following antibodies were used: FAK (BD Biosciences 610087), MBD2 (Sigma M7318), Hic-5 (BD Biosciences 611164), androgen receptor (Novus Biologicals NBP2-67497).

## Dextran exclusion assay
Twenty, 40, and 65–85 kDa dextrans (Sigma, FD20S/FD40S/T1162) were dissolved in 'Transport Buffer (TB)' (20 mM HEPES (pH 7.3), 110 mM potassium acetate, 5 mM sodium acetate, 1 mM EGTA, and 2 mM DTT) at 10 mg/ml. Degradation products were removed using 10 and 30 kDa-cutoff size exclusion columns (Millipore UFC501096/UFC503096). Dextran stock solutions were diluted in TB 1:50 to 1:200 prior to use. C2C12 cells seeded in a chambered cell culture slides were incubated with TB supplemented with 20 µg/ml digitonin for 5 min at room temperature, washed with TB (without digitonin) three times, and added dextrans. After 7.5 min at 37 °C, images were acquired using a Leica SP8 confocal microscope equipped with a 63 x oil-immersion objective. Images were cropped and pseudocolored using FIJI.

## mRNA-sequencing

One to 2 million C2C12 cells were lysed in 1 ml TRIzol. 0.4 ml chloroform was added and vigorously shaken for RNA extraction. The aqueous phase was transferred to a fresh tube, and one volume of 70% ethanol was added dropwise while vortexing at the lowest speed at room temperature. The mixture was purified using RNeasy Mini kit (Qiagen) to yield several hundred ng/µL total RNA. mRNA-sequencing was performed in paired-end 150 base pair mode on the Illumina NovaSeq 6000 platform. Paired-end fragments were mapped to mm10, filtered, and assembled into transcripts using HISAT2 (*Kim et al., 2019*), SAMtools (*Li et al., 2009*), and StringTie (*Pertea et al., 2015*). Differential expression was evaluated using DESeq2 (*Love et al., 2014*), and further analyzed with in-house python scripts.

## Mass spectrometry

Samples were precipitated by methanol/chloroform and redissolved in 8 M urea/100 mM TEAB, pH 8.5. Proteins were reduced with 5 mM tris(2-carboxyethyl)phosphine hydrochloride (TCEP, Sigma-Aldrich) and alkylated with 10 mM chloroacetamide (Sigma-Aldrich). Proteins were digested overnight at 37 °C in 2 M urea/100 mM TEAB, pH 8.5, with trypsin (Promega). The digested peptides were labeled with 10-plex TMT (Thermo product 90309, lot VI307195B), pooled samples were fractionated by basic reversed phase (Thermo 84868).

The TMT-labeled lysate samples were analyzed on a Orbitrap Eclipse mass spectrometer (Thermo). Samples were injected directly onto a 25 cm, 100 µm ID column packed with BEH 1.7 µm C18 resin (Waters). Samples were separated at a flow rate of 300 nL/min on an EasynLC 1200 (Thermo). Buffer A and B were 0.1% formic acid in water and 90% acetonitrile, respectively. A gradient of 1–25% B over 100 min, an increase to 40% B over 20 min, an increase to 100% B over 10 min and held at 100% B for a 10 min was used for a 140 min total run time.

Peptides were eluted directly from the tip of the column and nanosprayed directly into the mass spectrometer by application of 2.5 kV voltage at the back of the column. The Eclipse was operated in a data dependent mode. Full MS1 scans were collected in the Orbitrap at 120 k resolution. The cycle time was set to 3 s, and within this 3 s the most abundant ions per scan were selected for CID MS/MS in the ion trap. MS3 analysis with multinotch isolation (SPS3) was utilized for detection of TMT reporter ions at 60 k resolution (*McAlister et al., 2014*). Monoisotopic precursor selection was enabled and dynamic exclusion was used with exclusion duration of 60 s.

Protein and peptide identification were done with Integrated Proteomics Pipeline – IP2 (Integrated Proteomics Applications). Tandem mass spectra were extracted from raw files using RawConverter (*He et al., 2015*) and searched with ProLuCID (*Xu et al., 2015*) against Uniprot mouse database. The search space included all fully-tryptic and half-tryptic peptide candidates. Carbamidomethylation on cysteine and TMT on lysine and peptide N-term were considered as static modifications. Data was searched with 50 ppm precursor ion tolerance and 600 ppm fragment ion tolerance. Identified proteins were filtered to using DTASelect (*Tabb et al., 2002*) and utilizing a target-decoy database search strategy to control the false discovery rate to 1% at the protein level (*Peng et al., 2003*). Quantitative analysis of TMT was done with Census (*Park et al., 2014*) filtering reporter ions with 20 ppm mass tolerance and 0.6 isobaric purity filter.

## Quantification and statistical analysis

Statistical analyses for *Figures 1E, 2D and G*, *Figure 1—figure supplement 3G*, and *Figure 5—figure supplement 1G* were performed using GraphPad Prism.

## Acknowledgements

We thank the members of the Hetzer laboratory, Tony Hunter (Salk), Lorenzo Puri (Sanford Burnham Prebys), and Jongmin Kim (Massachusetts General Hospital) for the critical reading of the manuscript; Kenneth Diffenderfer and Aimee Pankonin (Stem Cell Core at the Salk Institute) for help with neurogenesis; Carol Marchetto and Fred Gage (Salk) for providing H9 embryonic stem cells; Lorenzo Puri, Alexandra Sacco, and Luca Caputo (Sanford Burnham Prebys) for helpful discussions and sharing mouse primary myoblasts. This work was supported by a Glenn Foundation for Medical Research Postdoctoral Fellowship in Aging Research (UHC), the NOMIS foundation (MWH), and the

National Institutes of Health (R01 NS096786 to MWH and K01 AR080828 to UHC). This work was also supported by the Mass Spectrometry Core of the Salk Institute with funding from NIH-NCI CCSG: P30 014195 and the Helmsley Center for Genomic Medicine. We thank Jolene Diedrich and Antonio Pinto for technical support.

## Additional information

### Funding

| Funder | Grant reference number | Author |
|---|---|---|
| National Institute of Arthritis and Musculoskeletal and Skin Diseases | K01 AR080828 | Ukrae H Cho |
| National Institute of Neurological Disorders and Stroke | R01 NS096786 | Martin W Hetzer |
| Glenn Foundation for Medical Research | Postdoctoral Fellowship | Ukrae H Cho |
| NOMIS Stiftung | | Martin W Hetzer |

The funders had no role in study design, data collection and interpretation, or the decision to submit the work for publication.

### Author contributions

Ukrae H Cho, Conceptualization, Formal analysis, Funding acquisition, Investigation, Visualization, Methodology, Writing – original draft, Writing – review and editing; Martin W Hetzer, Supervision, Funding acquisition, Project administration, Writing – review and editing

### Author ORCIDs
Ukrae H Cho ![ORCID] https://orcid.org/0000-0001-8796-810X
Martin W Hetzer ![ORCID] https://orcid.org/0000-0002-2111-992X

Reviewer #1 (Public Review): https://doi.org/10.7554/eLife.89066.2.sa1
Reviewer #2 (Public Review): https://doi.org/10.7554/eLife.89066.2.sa2

## Additional files

### Supplementary files
• Supplementary file 1. Raw data for the mass spectrometry experiment described in *Figures 3 and 4*.

• Supplementary file 2. Antibodies, plasmids, RNA probes, and chemicals used in this study. (a) Primary antibodies used in this study (b) Secondary antibodies used in this study (c) Plasmids used in this study (d) RNA FISH probes used in the study (e) Chemicals used in this study

• MDAR checklist

• Source data 1. Ponceau S staining and loading control for immunoblots shown in this study.

### Data availability
RNA-sequencing data have been uploaded to the Gene Expression Omnibus (GEO) database (NCBI) under accession number GSE183521.

The following dataset was generated:

| Author(s) | Year | Dataset title | Dataset URL | Database and Identifier |
|---|---|---|---|---|
| Cho UH, Hetzer MW | 2021 | mRNA-sequencing of C2C12 myoblasts, myotubes, and reserve cells | https://www.ncbi.nlm.nih.gov/geo/query/acc.cgi?acc=GSE183521 | NCBI Gene Expression Omnibus, GSE183521 |

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
