## [Editor Report · eLife assessment]

This **important** study outlines a new role for caspases during cellular differentiation. The methodology used is **convincing** and state-of-the-art. The newly discovered cellular cascade described here uncovers that caspases can achieve high substrate specificity during differentiation. As such, the work will be of broad interest to cell biologists.

---

## [Referee Report · Reviewer #1 (Public Review)]

This manuscript describes the transient proteolysis of several Nups during myogenesis due to activation of caspase 3, and how this "trimming" leads to defects in nuclear export. The authors show the NPC-related course of events during cellular differentiation and suggest mechanistic insights into exactly why this limited proteolysis is needed for myogenesis. In addition, the authors introduce a novel concept for caspase cellular function that might be worth investigating in the future. Overall, the authors present an elegant and interesting piece of work, performed at the usual superb quality of this group, and indeed the figures throughout the manuscript clearly show a very high level of experimental expertise.

---

## [Referee Report · Reviewer #2 (Public Review)]

Cho and Hetzer provide evidence that nuclear pore complexes (NPCs) are "trimmed" by caspases as a key element of muscle (and other) differentiation programs. Overall, the data are of high quality and are well presented. There is an interesting mechanism demonstrated whereby nuclear and cytosolically-oriented nups are specifically degraded from the NPC (fragments are sometimes associated with the NPCs), which leads to a specific inhibition of nuclear export. A highlight is a quantitative proteomic analysis of nuclear fractions that nicely demonstrates the change in the nuclear proteome upon NPC trimming, which includes elevated levels of many NES-containing factors. An important control is that these nuclear proteome changes don't occur when caspases are inhibited. These data are valuable although they fall short in demonstrating that NPC trimming is actually required for the execution of the differentiation program. It is recognized, however, that editing several nup genes at several sites to prevent caspase recognition would be extremely challenging and unfeasible, thus ultimately this does not detract from the significance of the findings. Indeed, there is a new broadly impactful concept being introduced - that caspases need not be destructive but they can be productively utilized to contribute to cell fate decisions.